# Informed Consent for Newborn Genomic Screening: Interest-Holder Perspectives on Dynamic Consent in an Evolving Landscape

**DOI:** 10.3390/ijns11020041

**Published:** 2025-05-28

**Authors:** Marina Okamura, Emma Minchin, Carolyn Mazariego, Jolyn Hersch, Natalie Taylor, Ilona Juraskova

**Affiliations:** 1School of Psychology, The University of Sydney, Sydney, NSW 2006, Australia; marina.okamura@sydney.edu.au (M.O.); emma.minchin@sydney.edu.au (E.M.); jolyn.hersch@sydney.edu.au (J.H.); 2Implementation to Impact (i2i), School of Population Health, University of New South Wales, Sydney, NSW 2052, Australia; c.mazariego@unsw.edu.au (C.M.); natalie.taylor@unsw.edu.au (N.T.); 3School of Public Health, The University of Sydney, Sydney, NSW 2006, Australia

**Keywords:** newborn screening, informed consent, genomic testing, dynamic consent platform, qualitative research

## Abstract

Newborn Bloodspot Screening (NBS) has significantly advanced early disease detection, preventing severe disability and infant mortality. The anticipated integration of genomic technologies into NBS (gNBS) promises earlier diagnosis and targeted treatments. However, it also introduces complexities that necessitate enhanced consent processes. Dynamic Consent Platforms (DCPs), with their layered information and modifiable preferences, may fulfil this rapidly evolving need. This qualitative study explored NBS and genomic interest-holder perspectives on (i) challenges in obtaining informed consent within the current and genomic NBS contexts, and (ii) the acceptability, feasibility, and utility of DCPs for genomics. Sixteen key interest-holders involved in NBS/genomic consent (midwives, genetic counsellors, geneticists, researchers, pathologist, consumer advocate) completed a semi-structured interview. Thematic analysis identified four main themes: (i) looking towards genomic expansions, (ii) systemic issues, (iii) genomic consent information, and (iv) Dynamic Consent Platforms. Participants emphasised revising the timing of consent processes and standardising consent training for clinicians. A nationally standardised DCP was perceived as valuable for addressing consent challenges within gNBS; however, concerns were raised regarding accessibility of online resources for vulnerable populations and integrating DCPs into healthcare systems. Recommendations for future research and clinical implications in this evolving field are discussed.

## 1. Introduction

Newborn Bloodspot Screening (NBS) is a highly successful public health program in Australia, with a 99% nationwide uptake, offering newborns timely screening for numerous serious or life-threatening yet treatable conditions [1]. For the past three decades, standard NBS (stdNBS) services have maintained a relatively consistent approach, utilising biochemical technology and screening for a limited number of disorders [2]. Integration of genomic medicine within the NBS could allow targeting of numerous rare genetic conditions not identifiable in pre-symptomatic infants under the current program [3], enabling screening for hundreds of rare diseases with variable expressivity, at a lower cost per disease [4]. Several international clinical trials are piloting the feasibility of integrating genomic sequencing into NBS [5]. Notably, the GUARDIAN study in the United States demonstrated the real-world feasibility of implementing genomic sequencing within a state NBS program, with high levels of parental acceptance [6]. In the context of the evolving and complex landscape of genomics, obtaining informed consent for testing is critical.

In Australia, there are five independent NBS programs amongst the eight states and territories, with variation in the comprehensiveness of consent practices for testing, data storage, and research amongst the states and programs [1,7]. Expanding the screening program to include genomic NBS (gNBS) would require additional consent beyond current practices. Without adequate consent, parents may struggle to fully understand the purpose and implications of genomic testing, including the psychological burden of uncertain results (e.g., variants of unknown significance), potential privacy risks associated with extensive data storage, and the stress of receiving unexpected or complex health information [8]. Genomic sequencing substantially increases the volume and complexity of data generated, and the public health context of NBS programs presents unique challenges in ensuring that consent remains informed, meaningful, and appropriate for this population-level setting [9]. Therefore, new consent pathways and resources need to be explored to support Australian clinicians and parents if genomic-based technologies are to be routinely introduced within the NBS.

With advancements in digital technology, Dynamic Consent Platforms (DCPs) are emerging in genomics to facilitate complex consent processes for clinical research [10]. They enable participants to consent to research studies and communicate preferences in real time regarding research involvement, result return, data use and storage, mode of communication and consent for future studies [11]. DCPs meet the highest ethical and legal consent standards, include tailored information delivery through a range of modalities (e.g., videos, audio, text) and maintain electronic documentation of engagement in research activities [12].

An existing Australian prototype web-based dynamic platform, CTRL (“Control”), was developed for genomic research studies [13], providing granular consent options. Informed by semi-structured interviews, the platform’s acceptability has been demonstrated as comparable to traditional paper resources. Building on CTRL, the CoGenT (COnsent in GENomic Testing) resource was developed to enhance the quality of decision-making and consent within the cancer genomic research context [14]. This was in response to qualitative findings indicating that cancer patients often left the genomic consent process with limited understanding, inflated expectations, confusion, and evolving information needs [15]. The CoGenT DCP integrates questions and answers (Q&As) and a user-friendly interface, designed to support accessibility for patients of varying health literacy levels. The Q&As, developed as an extension of traditional question prompt lists, encourage patient/clinician/researcher dialogue, allowing patients to express concerns and gather information tailored to their needs [14]. Although DCPs represent progress towards improved recruitment, retention and engagement of participants for genomic research, they are not directly transferable to general consent used for population screening as the platform is context-specific [14,16]. However, they could show future applications if population consent procedures evolve alongside the integration of genomics in newborn screening. Further exploration of the potential applicability of DCPs in this space is therefore warranted.

This qualitative study aimed to explore interest-holder views on current consent processes and those related to the prospective integration of genomic testing into the Australian NBS program. Interest-holders’ perceptions of the potential acceptability, feasibility, and utility of an online DCP such as CoGenT were also examined. Specifically, the study explored perceived barriers and facilitators to implementing a national DCP in NBS programs and potential modifications required for CoGenT to be applicable to a population-level program like NBS. For the purpose of this study, the term ‘interest-holders’ describes groups directly involved in health-related decisions due to the colonial connotations associated with the term ‘stakeholders’ [17].

## 2. Materials and Methods

Ethical approval was obtained from The University of Sydney Human Research Ethics Committee (2024/HE000561).

### 2.1. Participants

To capture a broad range of perspectives from both NBS clinicians and genomic experts, eligible participants included key interest-holders such as geneticists, genetic counsellors, midwives/nurses, researchers, consumer advocates, pathologists and operation/project managers. Additional eligibility criteria included being adults with access to a computer and sufficient English proficiency to complete the survey and interview. The sample size was guided by information power [18], with ongoing assessments made throughout data collection based on participants’ familiarity with consent in NBS and/or consent for genomic testing within varying Australian NBS programs. Recruitment ceased once data reached sufficient variability among interest-holders, which was achieved in this study with 16 participants.

### 2.2. Procedure

Participants were recruited during a gNBS-focused interest-holder workshop and via wider purposive sampling. Prospective participants expressed interest via a QR code and were sent an email with a Qualtrics study link which included the information sheet, electronic consent form and a brief survey. Upon receiving the survey responses, participants were emailed a Zoom interview invitation and a link to a three-minute YouTube video introducing the CoGenT prototype (Available online: https://www.youtube.com/watch?v=Gff74Ff-U0s (accessed on 1 March 2025)), which they were asked to watch before their interview.

### 2.3. Measures and Materials

Participants completed an 8-item demographic questionnaire capturing gender, highest level of education, occupation, employment length, work setting and workplace location, involvement in consent, and experience with genomic testing.

The development of the semi-structured interview guide was informed by the Consolidated Framework for Implementation Research (CFIR 2.0) [19] with iterative input from two experts in NBS genomics. Open-ended questions and prompts were used to facilitate comprehensive exploration of topics and sharing of unique perspectives [20]. Participants were initially prompted to discuss their perspectives on NBS consent and relevant changes if gNBS was to be introduced routinely, reflecting on their relevant work experience. The feasibility and appropriateness of DCPs within NBS, and CoGenT specifically, were also explored. The interview guide was tailored to specific interest-holder professions (e.g., genetic counsellors, midwives, etc.), to maximise data richness and quality. This guide is provided in Appendix A.

### 2.4. Data Analysis

Questionnaire responses were analysed descriptively. Interviews were transcribed verbatim, with transcripts de-identified. Thematic template analysis was used to analyse interview data. This approach employs a flexible coding template and an extensive hierarchical mapping of codes that facilitate the iterative refinement of data, thus allowing for a rich exploration of interest-holder perspectives [21]. Following data familiarisation, two researchers (MO, EM) coded six initial transcripts, selected to represent a range of interest-holder professions. An initial coding template was collaboratively developed by MO, EM and IJ, organising codes into preliminary themes and subthemes. This template was applied to the remaining transcripts with the aid of NVivo 14 software [22]. The template was further refined after all transcripts had been coded to reflect the final themes and sub-themes, which guided the write-up of the findings. Weekly research meetings (MO, EM, and IJ) were held to iteratively clarify and prioritise major themes, resulting in further refinement of the final coding template. Further improvements to the final template were made after these discussions. To ensure methodological rigour, we followed the four quality assurance techniques outlined by King and colleagues [23]: (i) independent coding of a portion of the transcripts (n = 6 out of 16) by two researchers (MO, EM) during the preliminary coding stage, (ii) thick description, supported by researchers’ reflexive statements acknowledging preconceptions and subjectivity prior to data collection, (iii) audit trails, a detailed step-by-step account of the methods documented and reviewed during weekly research meetings, and (iv) respondent feedback (member checking), where five participants reviewed and confirmed that a summary of the findings accurately reflected their experiences.

## 3. Results

Of the 24 eligible interest-holders who expressed interest in the study, 16 completed the interview, 7 were uncontactable and 1 withdrew (67% response rate).

Participant demographic and clinical characteristics are summarised in Table 1. Most participants were female (n = 13). Interest-holders’ occupations varied, although most were genetic counsellors (n = 5) or midwives (n = 5). On average, participants had 16.6 years of work experience in an NBS- and/or genomics-related role (range: 3–43). Most had postgraduate training (n = 11) and worked in the state of Queensland (n = 12).

### 3.1. Qualitative Findings

Four overarching themes and ten sub-themes were identified. The main themes were (1) Looking Towards Genomic Expansions, (2) Systemic Issues, (3) Genomic Consent Information, and (4) Dynamic Consent Platform (Figure 1). Findings that were related to stdNBS versus gNBS differences are explicitly noted; otherwise, shared elements are discussed.

### 3.2. Theme 1: Looking Towards Genomic Expansions

All participants emphasised the need to revise consent processes if genomics were to be integrated into routine screening (gNBS). They expressed uncertainty about the implications of this expansion, questioning whether the potential benefits would outweigh the possible harms, particularly if consent processes are insufficient:

*“*[NBS] *is lifesaving in a lot of cases and can prevent pretty significant disability… the risk of false positives or the impact on families… those things need to be quite clearly articulated.”*[P38, genomics project coordinator]

Interest-holders with genomic expertise were more likely to emphasise the “*risk if consent is not done adequately”* in the context of gNBS. In the event that consent processes become more rigorous to accommodate genomic expansion, most participants predicted only a slight decline in screening rates. They noted that such concerns should not hinder efforts to enhance the consent process, particularly considering the additional ethical and informational needs posed by genomic sequencing. However, a few participants expressed concern that increasing the rigour of the consent process could jeopardise screening uptake, potentially leading to delayed diagnosis and increased mortality among affected infants:


*“We’ve got to be very, very careful. We’ve got to not stop people from having the initial screen. We can save babies within seven days of their lives.”*
[P42, midwife]

Most interest-holders also voiced significant concerns about consent delivery as the general population’s unfamiliarity with stdNBS was noted, further complicating efforts to inform parents about genomics during a time of heightened stress and exhaustion:


*“That’s the problem now, [parents] don’t have any [consent] information and then all of a sudden, the baby’s born…that’s fine with biochemical [tests] because you’re actually trying to see if they’ve got the condition. But once you move into genomics, they may not have the condition there and then.”*
[P28, pathology operations manager]

### 3.3. Theme 2: ‘Cracks of the Health System’—Systemic Issues

All interest-holders described current consent information and procedures for stdNBS as largely suboptimal, with most acknowledging that the current health system is ill-equipped to integrate gNBS into routine NBS.

#### 3.3.1. Timing of Consent

Participants reflected that asking parents to consent to stdNBS within 72 h post-birth (i.e., at the time of the heel prick) is poorly timed, as they are *“completely sleep deprived, slightly traumatised and overwhelmed”* [P48, researcher]. Interest-holders across states and professions questioned whether this timing compromises informed consent, overwhelmingly deeming it inappropriate. Instead, most interest-holders proposed a *“two-part consent process”*, suggesting that the ideal time to obtain consent for stdNBS would be during the antenatal period, with consent reconfirmed at the time of the heel prick. This approach was strongly recommended, particularly with the potential introduction of gNBS:

*“Prenatal* [consent introduction] *…when you have time to actually absorb it and understand what it’s all about and have a pathway to ask those questions.”*[P24, consumer advocate]

Whilst midwives agreed that obtaining consent antenatally would be beneficial, many emphasised the importance and challenge of *‘capturing everyone’* due to variations in antenatal care models across Australia.

#### 3.3.2. Supporting Health Professionals in Consent Delivery

Most interest-holders stressed that consent should *“always be supported by a discussion with a healthcare professional”* [P38, researcher], particularly with gNBS introduction. However, some raised concerns about workforce capacity and potential impacts of this expansion on front-line workers:

*“It’s just not feasible to have* [health professionals] *having these extensive consent conversations with every single person who’s pregnant… the workforce is so short-staffed.”*[P34, genetic counsellor]

All midwives stressed that there is currently no formal training for delivering consent and that midwives therefore simply *“watch and learn”* from other health professionals. They were also unclear how to instigate and implement improvements in consent delivery:


*“That would mean a whole different lot of training. You’ve got thousands of midwives that do this… so how do you implement that? How do you monitor it? Who oversees it?”*
[P42, midwife]

Queensland midwives reported that since consent delivery is not currently standardised, the quality and content of any consent interaction largely depends on the experience and knowledge of each health professional, resulting in significant variation:

*“Every* [midwife] *has their own varying spiel…it depends on their understanding of the newborn screening, because there isn’t a blanket consent process.”*[P36, midwife]

Some midwives discussed the need for an *‘education period’* if gNBS is introduced while acknowledging that *“having some sort of script”* might assist health professionals to safely obtain informed consent as competing demands leave “*never enough time for education”*.

Most interest-holders strongly advocated for standardising consent procedures nationally, especially if gNBS is introduced. Some described the current system as a ‘*postcode lottery’* due to its fragmentation across regions within Australia:


*“We’re not a big enough population to be having eight different versions of a consent process for a national population health initiative.”*
[P46, researcher]

### 3.4. Theme 3: Genomic Consent Information

All interest-holders discussed the nature of information necessary for facilitating gNBS informed consent, which was largely dependent on their varied assumptions about genomic expansions and test implications.

#### 3.4.1. Content of Consent Information

Interest-holders suggested that parents should receive information regarding the test’s purpose, the heel prick procedure, potential benefits and risks, likelihood of a positive result, care plans for a positive result, data storage and sharing, potential impacts on insurance and access to services:

*“There are two parts: what they’re going to do with the test* [purpose]*, but it’s also consent to how they’re going to collect that blood as well* [procedure]*.”*[P42, midwife]

With the expansion to genomics, participants noted the importance of conveying essential information without making it too *‘cumbersome’* or detailed*,* pointing to the large amount of information parents commonly receive during the antenatal period. Some noted that tailoring and layering the information would prevent parents feeling overwhelmed.


*“The balance is having the different levels of information there for people that want it but keeping it relatively simple for those that don’t want that extra level of information.”*
[P22, genetic counsellor]

Some interest-holders with experience in developing consent for research suggested that since NBS is a public health initiative, consent for gNBS could be *‘test agnostic*’, meaning that detailed technical information about genomic sequencing may not be essential:

*“The most important things that people need to understand is the outcomes, the benefits and the risks… whether it’s biochemical or genomic* [technology]*.”*[P38, researcher]

#### 3.4.2. Understanding Implications

Some participants emphasised that parents currently consent to NBS often without fully understanding the screening implications, leaving them feeling *“shocked and traumatised in some cases, when they do get that positive result”* [P38, researcher]. Most interest-holders acknowledged that since there are varying information-seeking needs, additional resources should be available to support parents’ knowledge during the consent process:


*“Yes, it’s still going to be a shock [if the test comes back positive] but at least they know what the test was.”*
[P34, genetic counsellor]

Several also noted that although parents have the right to decline screening, consent information must clearly outline the consequences of such a decision:


*“So, what if you don’t have the test? What are the implications?”*
[P38, researcher]

### 3.5. Theme 4: Dynamic Consent Platform

All interest-holders expressed positive views about introducing a DCP for NBS, particularly in the context of gNBS; however, they also highlighted that the newborn context would necessitate specific considerations.

#### 3.5.1. Consent Platform

Most interest-holders agreed that individuals currently in their reproductive years are typically *‘tech savvy’*, suggesting an online modality would be more suitable than current paper forms since *“the parents of today do everything online.”* [P24, consumer advocate]


*“We put a man on the moon 60 years ago, and we still don’t have an online genomic consent form.”*
[P20, genetic counsellor]

All interest-holders perceived the CoGenT platform to be *‘very user friendly’*. Most emphasised and valued the functions of providing layered consent information and modifying consent preferences, to accommodate varying user needs:

*“There are clear advantages* [of CoGenT] *around availability and access of information, ability to: launch to other resources, capture and track data, and have a granularity of consent for some of those optional opt ins.”*[P50, geneticist]

Some interest-holders commented how the Q&A feature integrated within the DCP could offer parents a useful educational resource, as is structured in a layered format; thus, they perceived the potential to facilitate informed decision-making:


*“It’s pitched at the level that the user would like to go to, so if they want more information, a lot of it is there… but if they want to keep the information fairly simple, then, it allows that as well.”*
[P22, genetic counsellor]

Many interest-holders noted that a DCP like CoGenT would also benefit health professionals, reducing the pressure on them to fully understand genomics.

*“*[The DCP] *gives healthcare providers something to say: ‘you can go home and go through this’. It’s very comprehensive information… as midwives, we’re not geneticists.”*[P60, midwife]

However, the need for adjunctive discussions with health professionals was emphasised to ensure patient understanding.

#### 3.5.2. gNBS-Specific Implementation Considerations of DCP

All interest-holders agreed that implementing a DCP in the newborn context necessitates specific considerations, particularly since CoGenT was designed for oncology.

##### Consumer Access and Potential Inequalities

Interest-holders from various professions noted that a DCP would only be feasible if implemented antenatally, highlighting the need for ongoing access for parents. The primary concern among interest-holders was effectively communicating consent information to pregnant mothers with varying levels of general literacy, health literacy, digital literacy, or internet access:


*“There are people who are not digital…so you will need to have a non-digital and maybe non-dynamic consent component available.”*
[P48, geneticist]

A few interest-holders also noted that genomics could be ‘*taboo’* in some cultures, raising important considerations for culturally sensitive consent delivery:

*“How* [communities] *need to be supported in this process…if you’re aiming for everyone to be fully informed you need to go to community and find these things out”*[P38, researcher]

However, while acknowledging potential inequalities in accessing technology, a midwife addressed that technological devices are available for patient use in hospitals:

*“Most units have an iPad or something that* [midwives could give to parents] *or like social work* [staff] *would be able to assist those who may not have access to that.”*[P36, midwife]

##### Access for Health Professionals

Some interest-holders reported that a DCP will be ineffective unless various health professionals can readily access patients’ consent information. They emphasised the platform must be securely integrated with existing systems, including children’s health records and pathology departments, to ensure seamless functionality:

*“If* [the DCP data] *is not accessible to midwives, maternity nurses, even paediatric nurses in ED departments, your GPs… If* [I], *as a clinician at 48 h* [after a baby is born]*, do not have access to that consent, what’s the point of doing it?”*[P42, midwife]

Interest-holders discussed workload concerns, noting that a DCP’s acceptability hinges on its ability to reduce health professionals’ workloads. Concerns were expressed about the workflow of midwives if the DCP is introduced antenatally and parents have not provided consent before birth.

*“If it adds to the time for a midwife to get the newborn screen done, then* [the DCP] *is not going to be favourably received.”*[P20, genetic counsellor]

##### Practicality of Dynamic Consent

While most interest-holders broadly supported the use of dynamic consent in the newborn screening context, it was unclear if all participants fully understood how it differs from standard online consent forms. Those working in genetics or research demonstrated the clearest understanding and support for DCPs.

*“The positive side is that* [the DCP] *is dynamic, and that people can change their minds and update* [their preferences*] … with the paper* [consent] *is locked in time.”*[P48, geneticist]

Some interest-holders suggested the facilitation of eliciting parent preferences for testing of specific conditions could be advantageous. However, those more familiar with DCPs also questioned whether parents would fully grasp the implications of flexible preferences, noting that dynamic/changeable preferences should not be offered regarding specific conditions and tests:


*“It would be hard to give people options in terms of what they would be testing for—having to then select different samples with different tests and things, I think it just becomes a bit of a nightmare for the lab.”*
[P34, genetic counsellor]

## 4. Discussion

This study explored key interest-holder perspectives regarding informed consent for NBS in the context of potential genomic expansion and the feasibility and acceptability of a national digital consent resource in the newborn context. Four themes were identified, eliciting novel insights around perceived barriers and facilitators for nationwide consent developments in Australia, complementing emerging genomic literature in this area [24].

Study participants highlighted the importance of maintaining high screening rates and optimising the timing of consent processes if gNBS was to be routinely introduced. Promoting screening participation and fostering trust in the NBS program are essential for a successful public health initiative [1]. Interest-holders identified several barriers to obtaining informed consent, including inconsistent antenatal care models and varying consent procedures across independent state-driven stdNBS programs. Many perceived the current timing of consent as inadequate, indicating that challenges extend beyond standardising educational resources [1]. To address these concerns, participants endorsed a two-step consent process: an initial comprehensive *antenatal* consent discussion followed by consent re-confirmation at the time of the heel prick. Refining the timing of consent in this way may help parents better process key information, facilitate meaningful parent/clinician discussions and reinforce trust in the program [3,24].

The lack of standardised training for clinicians delivering NBS information and consent reflected in the current findings represents another barrier in obtaining informed consent. While midwives are clinically trained to perform the heel prick procedure, qualitative research indicates inadequate information delivery of informed consent in the current stdNBS [24]. Given the limited availability of genetic counsellors to provide individual consultations to all parents, additional professional development for midwives is recommended. Professional development has been associated with improved confidence and competence for clinicians communicating genomic information to the public [25]. However, insufficient time for training may hinder widespread midwifery access to professional development regarding gNBS consent [26]. Current study participants suggested introducing example wordings to assist clinicians delivering gNBS information, as the use of standardised scripts has been shown to be effective in integrating policy change into routine midwifery practice [27].

### 4.1. Exploring the Role of DCPs for gNBS

This is the first study to qualitatively explore the utility of a DCP in the newborn context, considering potential genomic expansions. Interest-holders perceived a communication resource like CoGenT as helpful in improving informed consent processes in this setting, addressing the key needs identified by the clinicians delivering the consent information to parents. The growing technological fluency among individuals of reproductive age and CoGenT’s design were two key facilitators underlying support for the tool. Specifically, CoGenT was perceived as facilitating informed consent and also likely to sustain uptake due to the layered information accommodating parents’ varying information needs [28] and its interactive format fostering antenatal discussions [19]. Previous findings from clinical trials indicate that question prompt lists, like CoGenT’s adapted Q&A format, foster patient/clinician exchanges, as they facilitate discussions tailored to individual needs [14,19].

The scope of the CoGenT platform was viewed positively; however, participants suggested simplification of the platform to address the unique context of the NBS program. The findings indicated that, as a national screening program, parents should not be allowed to selectively opt out of specific conditions. However, in line with emerging genomic literature, we found support for a refined DCP with content relating to preferences for data storage and ongoing research participation [29]. This endorsement aligns with the unprecedented potential of genomic technology and the current inability to anticipate the prospective use of stored samples [30]. As DCPs can evolve with changing ethical and legal consent requirements, dynamic consent resources like CoGenT may yield potential for the prospective gNBS setting [12,30].

The digital modality was identified as both a strength and potential barrier to the feasibility of implementing the platform. Participants raised concerns for parents who may not have access to technology/the internet. To address barriers, participants suggested providing a paper copy or offering iPads, which are already available for parents in hospitals and maternity centres, to access a DCP during antenatal and postnatal care. Future studies should assess the preferences and feasibility of various formats of NBS consent delivery.

It is important to consider overarching contextual factors that influence the acceptability of DCP implementation. The technological infrastructure was perceived as an important barrier along with organisational challenges, as adopting digital shifts in standard healthcare is often met with resistance [26]. This was also noted by Haas and colleagues [10], who identified that, to realise the full potential of DCPs, organisational challenges, such as assessing technical readiness to integrate online consent resources with electronic medical records, must be addressed.

While the present findings yield valuable insights into the current and foreseeable barriers to refining NBS consent practices and suggest potential solutions in preparation for genomic expansions, policy-level changes and implementation frameworks will be required to improve the healthcare system. The current study indicates that DCPs may be conditionally accepted in the newborn context, and this will be dependent upon how ‘dynamic’ preferences for consenting to gNBS are, as well as accessibility to Australian parents and how implementation could be incorporated into pre-existing healthcare systems. Participants acknowledged standardising consent throughout Australia as an important first step. Nevertheless, a DCP like CoGenT could be successfully implemented to concurrently resolve national inconsistency and prepare systems for genomic expansions. To address barriers of implementation, a study involving NBS and genomic experts could identify relevant content, and targeted and flexible strategies to ensure a simplified DCP can be incorporated into clinical workflows. Implementation mapping informed by CFIR 2.0 [19] may be used to maximise identified facilitators and provide guidance on how to operationalise strategies to overcome barriers for successful DCP implementation.

### 4.2. Study Limitations

This study has several limitations. The sample size (N = 16) was small, limiting information power and transferability. Most participants were based in Queensland, which may have introduced regional bias, particularly given the state’s known challenges in consent practices compared to others [7]. Additionally, some key interest-holder groups, such as consumer advocates and pathologists, were underrepresented. Although over a third of the transcripts were double-coded by two researchers, the remaining interviews were analysed by a single coder. While this may have introduced interpretive bias, thematic consistency was supported through regular team discussions. Importantly, we employed a range of established strategies to ensure rigour, credibility and trustworthiness, including reflexivity, memo writing, audit trails, and member checking—longstanding practices in qualitative inquiry that complement and, in some views, extend beyond the focus on inter-coder reliability in public health and medical literature [31]. Finally, the study did not include parents, particularly those who received positive or false-positive screening results. Their perspectives are essential for evaluating the real-world relevance and acceptability of Dynamic Consent Platforms in newborn screening and require future research focus.

## 5. Conclusions

The current findings highlight the need to improve consent processes for stdNBS in Australia. A national policy framework for NBS was published in 2019 [1], but an extension specific to consent guidelines for upcoming genomic expansions to the program is crucial. Dynamic Consent Platforms, such as CoGenT, show promise in addressing key consent challenges associated with genomic testing in a public newborn screening program. To support the ongoing success of NBS and in preparation for genomic expansions, interest-holder discussions of DCPs offer valuable insights for future research and clinical practice in this evolving field.

## Figures and Tables

**Figure 1 IJNS-11-00041-f001:**
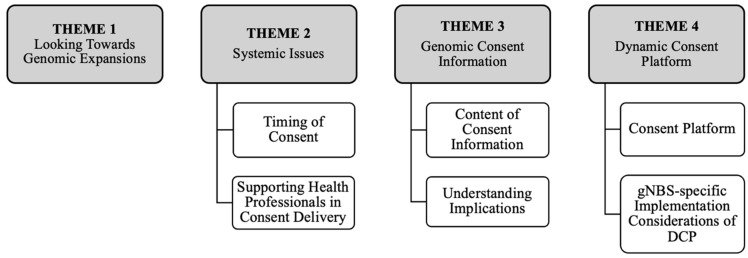
Diagrammatic representation of key themes and sub-themes.

**Table 1 IJNS-11-00041-t001:** Participant demographics and characteristics (N = 16).

		n (%)
Gender		
	Female	13 (81.3%)
	Male	3 (18.7%)
Occupation		
	Midwife	5 (31.3%)
	Genetic counsellor	5 (31.3%)
	Geneticist	2 (12.5%)
	Researcher	2 (12.5%)
	Consumer advocate	1 (6.3%)
	Pathology operations manager	1 (6.3%)
Years of working experience with NBS and/or genomics		
	<10	6 (37.5%)
	10–19	5 (31.3%)
	20+	5 (31.3%)
Work setting (s)		
	Public health system	11 (68.8%)
	Clinical research setting	2 (12.5%)
	Government	2 (12.5%)
	Patient support and advocacy	1 (6.3%)
	University/academia	1 (6.3%)
Work location		
	Queensland	12 (75%)
	New South Wales	2 (12.5%)
	Victoria	1 (6.3%)
	Western Australia	1 (6.3%)
Formalised training		
	Undergraduate university degree only	5 (31.2%)
	Postgraduate university degree	11 (68.8%)
Involvement in NBS/genomic consent processes		
	Yes	6 (37.5%)
Work experience with genomic testing		
	Yes	13 (81.3%)

## Data Availability

The de-identified interview data supporting the findings of this study can be made available on request from the corresponding author. The data are not publicly available due to privacy or ethical restrictions.

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
