# Peer review of "Informed Consent for Newborn Genomic Screening: Interest-Holder Perspectives on Dynamic Consent in an Evolving Landscape"

_2409-515X, 2025, doi:10.3390/ijns11020041_

Round 1

Reviewer 1 Report

Comments and Suggestions for Authors

The paper employs a quantitative semi-structured interview approach to explore perspectives on genomic newborn screening (gNBS) from 16 stakeholders. Overall, the study is clearly presented and maintains coherence with the topic of gNBS. However, the methods section could benefit from greater clarity and detail.

Methods:

Please briefly describe how the sample size was determined. According to the results, only 16 participants completed the study, while some withdrew or were uncontactable. It would be helpful to clarify whether 16 was the originally intended sample size.

Additionally, describe any measures taken to mitigate researcher bias during data collection and analysis.

Results:

Although the study identifies four themes and corresponding subthemes, the distribution of themes does not appear to be proportional to the depth or frequency of topic reflection. Further clarification or justification of the thematic structure would strengthen the results section.

Reviewer 2 Report

Comments and Suggestions for Authors

In line 45, the authors mentioned “Globally, informed consent is rarely obtained for stdNBS [6].” May the author elaborate on how the conclusion is drawn from reference 6? While it is understood that a significant number of NBS programmes, esp in most states of USA, list NBS as mandatory, with or without options of informed dissent, there are quite a number of programs in various parts of the world necessitate informed consent for biochemical NBS. For example, in Europe, “In a 51-country European study of newborn screening, participation was mandatory only in Italy while 64% of countries required consent, meaning that at least verbal consent was sought [25] (Personal communication (Loeber) Dec. 20, 2023). Indeed, in another 27-country study European newborn screening regulatory landscapes, only 11 required written informed consent” (Ref: https://www.nature.com/articles/s41431-024-01677-w )

In line 67-69; line 72 – 76; line 76 - 79, reference 13, a “manuscript under review” was cited. “Hersch, J., L. O'Hara, and I. Juraskova, Interventions to support patient decision making about taking part in health research: A 495 systematic review. Manuscript under review, 2024.” It would be difficult to review a citation of an article that is not available to the reviewer. Would the authors consider revising the citation ?

In line 170-171, “However, a few participants expressed concern that increasing the rigour of the consent process[22]”. Reference 22 (Scherer, N., et al., Mental health support for children and adolescents with hearing loss: scoping review. 2021.) was cited. May the author please elaborate on the citation?

For the practicality of dynamic consent platform (line 335 – 349), especially in a NBS context, did the interview covers specific issue related to NBS? E.g.

  • Transition of consent when the “newborn” transits from a minor to a major
  • Assuming the parents initially didn’t consent for gNBS and stdNBS, but later consent to screening in the dynamic platform, is there any practical consideration to such cases?

For the line 433-444, the authors had addressed several limitations of the study. However, some key interest-holders had not been included for the qualitative study, e.g. parents of newborns suffered from genetic disorders, and parents of newborns with false positive screening result. Their views would be key to the potential application of dynamic consent platform especially in a newborn screening context.

Round 2

Reviewer 1 Report

Comments and Suggestions for Authors

Thank you for making this revision.